# InTrathecal mORphine, traNsversus Abdominis Plane Block, and tramaDOl Infusion for Catheter-Related Bladder Discomfort in Patients Undergoing Robot-Assisted Laparoscopic Prostatectomy (TORNADO): A Pilot Prospective Controlled Study

**DOI:** 10.3390/jcm11082136

**Published:** 2022-04-11

**Authors:** Andrea Russo, Bruno Romanò, Domenico Papanice, Andrea Cataldo, Carlo Gandi, Luigi Vaccarella, Angelo Totaro, Emilio Sacco, Pierfrancesco Bassi, Paola Aceto, Liliana Sollazzi

**Affiliations:** 1Dipartimento di Scienze dell’Emergenza, Anestesiologiche e della Rianimazione, Fondazione Policlinico Universitario A. Gemelli IRCCS, 00168 Rome, Italy; andrea.russo@policlinicogemelli.it (A.R.); bruno.romano@policlinicogemelli.it (B.R.); domenico.papanice@policlinicogemelli.it (D.P.); andrea.cataldo@policlinicogemelli.it (A.C.); liliana.sollazzi@policlinicogemelli.it (L.S.); 2Università Cattolica del Sacro Cuore di Roma, 00135 Rome, Italy; angelo.totaro@policlinicogemelli.it (A.T.); pierfrancesco.bassi@policlinicogemelli.it (P.B.); 3Department of Urology, Fondazione Policlinico Universitario A. Gemelli IRCCS, 00168 Rome, Italy; carlogandi@gmail.com (C.G.); l.vaccarella87@gmail.com (L.V.)

**Keywords:** intrathecal morphine, transversus abdominis plane block, intravenous tramadol

## Abstract

Catheter-related bladder discomfort (CRBD), affecting surgical patients requiring large catheters, is often intolerable. In this prospective controlled study, we compared the efficacy of three analgesic approaches in the management of CRBD. Here, 33 patients undergoing robot-assisted laparoscopic prostatectomy (RALP) were allocated to the following three groups: intrathecal morphine (IM), transversus abdominis plane block (TAP), and tramadol intravenous infusion (TI). The primary outcome was CRBD assessed at admission in the recovery room (RR) (T0), and 1 h (T1), 12 h (T2), and 24 h (T3) after surgery. The secondary outcomes included the following: Aldrete score; postoperative pain, measured with a numerical rate scale (NRS) at T0, T1, T2, and T3; postoperative opioid consumption; and flatus. The patients of the IM group showed significantly lower CRBD values over time compared to the patients of the TI group (*p* = 0.006). Similarly, NRS values decreased significantly over time in patients receiving IM compared to patients treated with TI (*p* < 0.0001). Postoperative nausea and vomiting did not differ among the three groups. Postoperative opioid consumption was significantly lower in the IM group compared to the other two groups. Most patients of the IM group (9 of 11) had flatus on the first postoperative day. In conclusion, IM may prevent CRBD and reduce pain perception and postoperative opioid consumption and expedite bowel function recovery.

## 1. Introduction

The presence of a large-diameter urinary catheter may cause distress and pain for most surgical patients. Increased urinary frequency with or without signs of overactive bladder represents the manifestation of catheter-related bladder discomfort (CRBD) [1]. The incidence of CRBD in catheterized patients reaches 47–90% [1]. Male gender and the diameter of the catheter are the two most important risk factors for the development of CRBD [2]. The main mechanism behind CRBD is the activation of muscarinic receptors, which lead to involuntary contraction of the bladder [3]. Different drugs with anti-muscarinic and anticholinergic properties have been tested to prevent and treat this complex syndrome [4,5,6,7], which were measured using a four-point scoring classification—first described by Agarwal and coauthors [8]. Drugs and techniques used to treat perioperative pain are often used for this uncomfortable condition [2]. Tramadol is a centrally acting opioid analgesic with anti-muscarinic properties, which has been proven to be effective but is accompanied by side effects such as nausea and vomiting [9]. Recently, some authors documented the benefits of transversus abdominis plane block (TAP block) on postoperative pain scores for patients undergoing RALP, but without evaluating the CRBD [10]. Intrathecal morphine has been proven to be effective in postoperative pain control and bladder spasms [11].

Based on this evidence, we designed a prospective controlled study in order to compare the effects of intrathecal morphine, TAP block, and tramadol intravenous infusion on CRBD in patients undergoing RALP. We then explored the efficacy on postoperative pain control and the safety of these strategies in terms of their side effects and complications.

## 2. Materials and Methods

This single-center, prospective controlled study was approved by our Institutional Ethic Committee (ID 3236) and was registered on ClinicalTrial.gov. (no. NCT04814745). Written informed consent was obtained from each patient before the study. All adult patients scheduled for RALP were screened for enrolment. The exclusion criteria were the following: severe heart dysfunction (NYHA stages III–IV), end-stage renal disease, and neurological disorders. Patients were assigned to one of the following groups, according to both their preference and additional exclusion criteria:-Intravenous tramadol infusion (TI) group;-Transversus abdominis plane block (TAP) group: obesity (BMI ≥ 30 kg/m^2^);-Intrathecal morphine (IM) group: coagulation disorders and platelet dysfunction;

If one group was completed, the eligible patients were allowed to choose between the two other groups. The primary outcome was the presence of CRBD assessed at different times postoperatively, as follows: admission in the recovery room (RR) (T0), 1 h after admission in RR (T1), and 12 h (T2) and 24 h (T3) after surgery. The secondary outcomes were as follows: (a) Aldrete score in RR; (b) postoperative pain at T0, T1, T2, and T3; (c) adverse side effects, including nausea and vomiting (PONV); need for antiemetic, postoperative opioid consumption; (d) postoperative gastrointestinal recovery (i.e., flatus); and (e) desaturation episodes (SpO_2_ < 92%).

### 2.1. Anesthesia Protocol

All of the patients underwent standard monitoring: electrocardiogram, non-invasive arterial blood pressure, pulsoximetry, diuresis, and expiratory gas concentration. The depth of anesthesia was based on the bispectral index (BIS) value, which was kept between 40 and 60. Anesthesia was induced with fentanil 2 µg/kg and propofol 2 mg/kg, whilst tracheal intubation was facilitated by administration of rocuronium 0.6 mg/kg. Anesthesia was then maintained with Sevoflurane adjusted according to the BIS value (40–60). All patients were mechanically ventilated with tidal volume of 7 mL/kg and the respiratory rate was adjusted to maintain the carbon dioxide end-tidal between 35 and 40 mmHg. Rocuronium 0.15 mg/kg was then repeated in order to keep a deep neuromuscular block (Post Tetanic Count ≤ 2). Fentanyl (0.5 mcg/kg, before prostate removal) or morphine boluses (0.03 mg/kg, after prostate removal) were intraoperatively administered depending on heart rate and mean arterial pressure variations. The threshold for fentanyl dosing was based on drug data sheet (600 mcg). Balanced solutions were administered at 1–5 mL/kg/h intraoperatively and 1000 mL for 24 h postoperatively.

For the IM group, before general anesthesia, patients received spinal analgesia with 0.15 mg of morphine at the L3–L4 level, using a 25 Gauge Withacre needle.

For the TAP group, before extubation, patients received an abdominal wall block using a linear probe. Using a posterior approach, a 20 gauge 100 mm needle was inserted and 5 mL of 0.9% NaCl was injected to determine proper placement of the needle. Once the needle position was confirmed (between internal oblique and transversus muscle sheath), 20 mL of 0.2% ropivacaine was bilaterally injected.

For the TI group, after intubation, a 2 mL/h elastomeric pump with 400 mg tramadol in 48 mL of 0.9% NaCl solution, was started. For all patients, before extubation, 1 g of paracetamol and 30 mg of ketorolac were administered. In addition, 4 mg of dexamethasone and 10 mg of metoclopramide were used to prevent PONV.

Boluses of morphine (0.03 mg/kg; maximum dose of 10 mg) were used to treat postoperative pain in the recovery room (RR), while 100 mg of intravenous tramadol was the rescue dose therapy for pain control during ward stay, administered if the numeric rating scale (NRS) was ≥5. In addition, 8 mg of ondansetron was the rescue agent for the treatment of PONV in the ward. All of the patients received 1 g of paracetamol every 8 h for the first 24 h after surgery.

### 2.2. Data Collection and Measurements

Preoperative and intraoperative data were collected by one of the investigators, whereas postoperative data collection was undertaken by another investigator, who was not aware of the allocation group. Pain scores were assessed using NRS (from 0 with no pain to 10 with worst pain ever felt).

CRBD was classified according to severity, as follows: (0) no discomfort, (1) mild (reported by the patient only if asked), (2) moderate (expressed by the patient even if not asked and not accompanied by any particular behavior), and (3) severe (expressed by the patient and with specific behavior such as flailing limbs).

The Aldrete score was assessed in RR for determining if patients could be safely transferred to the ward. This scoring system was based on the following five items: (1) consciousness, (2) mobility, (3) breathing, (4) circulation, and (5) color. A score of 0–2 points was given for each parameter (with 0 indicating more serious conditions, 1 corresponding to an intermediate level, and 2 representing the restored functions). Patients were discharged from the RR if the Aldrete score was ≥9.

The following data were recorded intraoperatively: duration of surgery, amount of infused balanced solution, diuresis, blood loss, etilefrine use, intraoperative fentanyl consumption, and morphine use. Postoperatively, we measured the following: the Aldrete score at arrival and discharge from the RR; CRBD, NRS, and PONV at T0, T1, T2 (12 h after surgery), and T3 (24 h after surgery); and any desaturation episode (SpO_2_ < 92%) during RR stay and in the first 24 h after surgery. Postoperative morphine and tramadol consumption and the need for antiemetic therapy (ondansetron administration) were also registered.

### 2.3. Statistical Analysis

As foreseen by a rule of the thumb in the pilot studies [12], we decided to enroll 11 patients per group, allowing for a 10% dropout, as 30 patients would meet the threshold for a sufficiently precise estimate of the variance of CRBD change. Clinical and demographic characteristics were described using descriptive statistics. Quantitative variables were described using the mean and confidence intervals. The qualitative variables were summarized using absolute values and percentage. Demographic and intraoperative and postoperative variables were analyzed using one-way ANOVA or the chi-squared test. The primary outcome was achieved by applying ordinal logistic regression in order to estimate the effect of different analgesic treatments. The results are expressed in terms of regression coefficients, which estimate the difference between treatment groups in the log odds of being in the upper category of the dependent variable. For NRS and PONV, a multilevel linear regression and a multilevel logistic regression were performed, respectively. Multivariable modelling was applied to adjust for between-patient differences in preoperative and intraoperative characteristics. More specifically, we entered age, BMI, and ASA score, as well as all intraoperative variables, in the multivariable models. In all of the multilevel multivariable models, we used fixed effects for treatment groups, patients’ preoperative and intraoperative characteristics, and time, whereas we used random effects for the patients. We also entered an interaction term between treatment groups and time in order to assess whether the effect of treatment changed with time. The covariance matrix was left unstructured.

## 3. Results

The final sample included 33 patients. The patients’ characteristics and intraoperative data are shown in Table 1. No differences were found for the demographic variables.

The patients of the IM group showed significantly lower CRBD values over time compared with the patients of the TI group (*p* = 0.006), whereas CRBD did not vary across time in patients of the TAP group (*p* = 0.35). Intraoperative morphine and morphine administered in the recovery room were significantly associated with CRBD (*p* = 0.01); borderline significance was found for the association between surgery duration or blood loss and CRBD (*p* = 0.07; see Table 2).

As shown in Table 3, the NRS values decreased significantly over time in patients receiving IM compared with patients treated using TI (*p* < 0.0001). On the contrary, the trend of NRS values over time in patients who underwent TAP was similar to that of the patients of the TI group (*p* = 0.14). Intraoperative morphine and morphine administered in the recovery room were significantly associated with NRS (*p* = 0.0001).

Intraoperative fentanyl consumption was significantly lower in the IM group compared to the other two groups (*p* < 0.0001). Regarding postoperative rescue analgesia, none of the patients in the IM group required morphine during their RR stay; the number of patients requiring morphine in RR was 11 vs. 4 in the TI and TAP group, respectively (*p* < 0.0001). Similar findings were observed for tramadol use in the ward (*p* = 0.0004). Gastrointestinal recovery function, expressed as flatus, was significantly faster in the IM group compared with the other groups (*p* = 0.02).

PONV did not show significant changes across the three groups (TAP group vs. TI group, *p* = 0.09; IM group vs. TI group, *p* = 0.11).

Surgery variables (including surgery duration, crystalloids, diuresis, and blood loss), ondansetron administration, Aldrete scores, and hospital stay did not differ among the three groups (Table 1). None of the patients had pruritus or respiratory depression (SpO_2_ < 92%) in the postoperative period.

## 4. Discussion

In this study, we compared three analgesic techniques to prevent CRBD in patients undergoing RALP. The main finding of the present study was the greater benefit associated with intrathecal morphine, regarding CRBD treatment. CRBD symptoms vary among patients, from a burning sensation, agitation, and pain in the suprapubic area to urinary urgency. Several pharmacological studies for managing CRBD, including antiepileptic agents and antimuscarinic drug injections, have been reported. This syndrome also reduces the quality of recovery and prolongs hospital stay [6,13].

To our knowledge, Park JY et al. are the only authors who have investigated CRBD in the setting of the Robot-Assisted Laparoscopic Prostatectomy. They found that ketorolac administration was significantly associated with a reduced incidence of CRBD and lower pain scores [14]. In the present study, all patients received intravenous Ketorolac 30 mg before extubation. Furthermore, Park JY et al. assessed CRBD up to 6 h after surgery, while we investigated the effects on CRBD in POD-1, and 12 h and 24 h after surgery. During gynecological surgery, 1.5 mg/kg of intravenous tramadol was proven to be effective in the treatment of CRBD [15]. In previous studies conducted in other surgical settings, tramadol was used in the control group to test the efficacy of other therapeutic strategies for CRBD [15,16]. In our study population, tramadol represented the postoperative rescue analgesic during ward stay, and was requested only by patients who received TAP block and by those receiving continuous infusion to manage CRBD and/or postoperative pain. Li S et al. did not report the absolute number of subjects affected by nausea and vomiting, while they stated that these adverse effects were lower in the control group [15]. We did not find significant differences in PONV incidence among groups and in the need for ondansetron. 

In this study, TAP block was not superior to tramadol infusion in CRBD treatment. The rationale behind the inclusion of the TAP block as a comparative technique was based on the fact that when performing the posterior approach of the TAP block, as done in this study, local anesthetic should extend throughout posterior medial pathway to the paravertebral region from the T5 to L1 level, resulting in a transient paravertebral blockade. The paravertebral space contains spinal nerves, as well as the dorsal and ventral branches of the spinal roots, and the sympathetic chain [17]. Therefore, the application of anesthetics into this space may result in sensory and sympathetic block with consistent visceral analgesic effects.

The efficiency of intrathecal hydrophilic opioids was recently investigated in a meta-analysis that found a reduced intraoperative and postoperative opioid consumption, pain scores and hospitalization in abdominal surgery [18]. The opioid sparing effect was found when using intrathecal opioids in addition to paracetamol or NSAIDs, as most studies used these drugs in the multimodal analgesia protocol. We also administered these drugs before extubation in all patients. In the mentioned meta-analysis, the authors were not able to detect a difference in the incidence of nausea, while they documented a dose-dependent effect for pruritus in the range of 100–800 mcg of intrathecal morphine. In line with this statement, we did not register pruritus in the IT group.

Few studies have explored the impact of intrathecal morphine administration on bowel function. Levy BF et al. demonstrated that intrathecal morphine showed an earlier recovery of the bowel function than epidural and PCA regimens [19]. In contrast with these results, no difference was found between intrathecal morphine and PCA morphine for bowel recovery function during colorectal surgery [20,21]. In the present study, we found a significant reduction in time to flatus; this was probably due to the lower doses of intraoperative fentanyl and postoperative intravenous morphine and tramadol.

Respiratory depression represents a harmful but rare complication associated with intrathecal morphine administration. Intrathecal morphine can produce dose-related analgesia and respiratory depression [22,23,24]. Different definitions of respiratory depression exist in the literature, from a low respiratory rate with high arterial PCO2 values [25,26], low oxygen saturation with pulse oximetry [27], or an increased level of sedation [28]. In our patients, 0.15 mg of intrathecal morphine did not cause any of these signs of respiratory depression, as we did not observe low levels of SpO_2_ and the Aldrete score in the recovery room showed a normal level of consciousness. As subarachnoid morphine administration carries the risk of rare but serious complications, such as epidural hematoma, the risk–benefit balance should be evaluated in individual patients and be based on the concomitant need to counteract pain in a stronger manner.

The main limitation of this study is the risk of bias linked to both the non-randomized design and the lack of blinding, although the methodological rigor was maintained in all trial phases. Another limitation is the small sample size, which did not allow for an appropriate comparison of the frequency of side effects, such as PONV. Moreover, the investigators did not directly control the group allocation, which was mainly based on patient’s preference, with some exception due to exclusion criteria.

In conclusion, the spinal technique with morphine administration at a 0.15 mg dosage, when compared to the continuous intravenous infusion of tramadol, may prevent CRBD in patients undergoing RALP. Moreover, it reduced postoperative pain scores without producing adverse side effects. Based on this data, we can suppose that intrathecal morphine may be an efficient pharmacological option for preventing CRBD.

## Figures and Tables

**Table 1 jcm-11-02136-t001:** Main preoperative, intraoperative, and postoperative parameters in the patients from the three groups. Values are means (95% confidence intervals) or numbers.

	TI Group (*n* = 11)	TAP Group (*n* = 11)	IM Group (*n* = 11)	F or χ^2^	*p*
Age, years	66.6 (62.8–70.7)	66.9 (63.0–70.8)	66.7 (62.9–70.6)	0.005	0.99
BMI, kg/m^2^	26.2 (24.3–28.0)	26.4 (24.5–28.2)	26.9 (25.1–28.7)	0.18	0.84
ASA (I/II/III)	1/9/1	1/7/3	0/11/0	5.39	0.25
Surgery duration, min	181 (165–198)	189 (173–205)	163 (147–180)	2.66	0.09
Balanced solution, mL	945 (675–1215)	845 (575–1115)	527 (257–797)	2.73	0.08
Diuresis, mL	241 (179–303)	227 (165–289)	186 (125–248)	0.88	0.42
Blood loss, mL	84 (45–122)	121 (83–149)	68 (30–106)	2.11	0.14
Etilefrine	1/10	1/10	4/7	3.67	0.16
Fentanyl, mcg	359 (310–408)	322 (274–371)	168 (119–217)	18.11	<0.0001
I.O. Morphine (Yes/No)	11/0	4/7	0/11	22.73	<0.0001
P.O. Morphine (Yes/No)	7/4	5/6	0/11	10.21	0.006
Tramadol (Yes/No)	9/2	4/7	0/11	15.48	0.0004
PONV T0 (Yes/No)	0/11	1/10	2/9	2.20	0.33
PONV T1 (Yes/No)	2/9	2/9	2/9	0.00	1.00
PONV T2 (Yes/No)	3/8	2/9	1/10	1.22	0.54
PONV T3 (Yes/No)	3/8	0/11	1/10	3.98	0.14
Ondansetron, RR (Yes/No)	7/4	5/6	4/7	1.70	0.43
Ondanseton, W (Yes/No)	1/10	1/10	2/9	0.57	0.75
Flatus * (at 1/2/3 days)	0/5/6	0/7/4	9/2/0	8.03	0.02
Aldrete score 1	9.3 (8.7–9.8)	9.7 (9.4–10.0)	9.5 (9.3–9.7)	1.56	0.23
Aldrete score 2	9.6 (9.3–10.0)	9.7 (9.4–10.0)	9.8 (9.5–10.1)	0.43	0.65
Hospital stay (days)	3.9 (3.0–4.8)	3.7 (2.9–4.6)	3.9 (3.0–4.8)	0.06	0.93

BMI: body mass index; ASA: American Society of Anesthesiologists’ physical status classification; RR: recovery room; W: during ward stay; Aldrete score 1: at RR admission; Aldrete score 2: at RR discharge; TI group: tramadol intravenous infusion; TAP group: transversus abdominis plane Block; IM group: intrathecal morphine; T0: admission to RR; T1: 1 h after admission to RR; T2: 12 h after surgery; T3: 24 h after surgery; * flatus is shown as counted if present at the first, second, and third postoperative day.

**Table 2 jcm-11-02136-t002:** Results of the multilevel multivariable ordinal logistic regression for CRBD (fixed effects).

	Coefficient	95% CI	*p*
Group (Reference = TI group)			
TAP group	−1.51	−3.96; 0.94	0.23
IM group	−2.26	−5.39; 0.88	0.16
Time (hours)	0.002	−0.07; 0.08	0.96
Group per time			
TAP group	0.05	−0.05; 0.15	0.35
IM group	−0.24	−0.41; −0.07	0.006
Surgery duration (min)	−0.02	−0.05; 0.002	0.07
Blood loss (mL)	−0.01	−0.02; 0.0008	0.07
Intraoperative morphine (mg)	3.13	0.66; 5.61	0.01
Morphine in RR (mg)	2.07	0.41; 3.71	0.01

TI, tramadol infusion; TAP, transversus abdominis plane block; IM, intrathecal morphine; RR, recovery room; CI, confidence interval.

**Table 3 jcm-11-02136-t003:** Results of the multilevel multivariable linear regression for NRS (fixed effects).

	Coefficient	95% CI	*p*
Group (Reference = TI group)			
TAP group	−0.007	−0.72; 0.74	0.98
IM group	−0.55	−1.47; 0.37	0.24
Time (hours)	0.06	−0.03; 0.09	0.0001
Group per time			
TAP group	−0.03	−0.08; 0.01	0.14
IM group	−0.11	−0.15; −0.06	0.001
Intraoperative morphine (mg)	0.98	0.34; 1.62	0.0001
Morphine in RR (mg)	0.88	0.44; 1.33	0.0001

TI, tramadol infusion; TAP, transversus abdominis plane block; IM, intrathecal morphine; RR, recovery room; CI, confidence interval.

## Data Availability

Data presented in this study are available upon request from the corresponding authors.

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
