# Peer review of "InTrathecal mORphine, traNsversus Abdominis Plane Block, and tramaDOl Infusion for Catheter-Related Bladder Discomfort in Patients Undergoing Robot-Assisted Laparoscopic Prostatectomy (TORNADO): A Pilot Prospective Controlled Study"

_jcm, 2022, doi:10.3390/jcm11082136_

Round 1

Reviewer 1 Report

In this manuscript, the authors describe the significance of intrathecal morphine administration for urinary catheter-induced discomfort in patients undergoing robot-assisted laparoscopic prostatectomy. In general, the theme in the manuscript is a critical topic in our research field: anesthesiology and perioperative medicine. However, there are a few major concerns with this manuscript that need to be addressed by the authors.

1. As the authors stated in the Discussion section, the main concern in this research is the lack of randomization. Non-randomized setting causes the presence of selection bias. Describe any efforts in your research to reduce selection bias.

2. The authors showed the primary outcome, the chronological changes in the severity of bladder discomfort, in Figure 1. As the authors stated in the Methods section, the severity of the discomfort is an ordinal variable. Therefore, the authors should NOT use repeated measures ANOVA in the statistic. Also, the use of means of the ordinal variable is wrong. 

3. In addition, in modern statistics, the application of ANOVA is not appropriate for repeated measurements. Many researchers recommend the use of generalized linear mixed modeling for longitudinal data. Please carefully read two excellent reviews: Anesth Analg 2018;127:569 (PMID29905618) and Neuron 2022;110:21 (PMID34784504). Try to perform GLMM analysis again.  

Also, the reviewer recommends changing several minor points as described below.

1. Introduction. The explanation of the three treatment methods (IM, TAP, TI) is abrupt and confuses the reader a little. The author should explain them more carefully and in detail.

2. Statistical analysis. The authors should explain in more detail how the study size was determined. 

3. Results. To help the reader's understanding, the authors should round off data to the closest whole number. 

4. Spelling and grammatical errors, particularly in the use of capital letters, should be corrected. The authors should request an English proofreading service to check the paper. 

Author Response

Responses to Reviewer 1

In this manuscript, the authors describe the significance of intrathecal morphine administration for urinary catheter-induced discomfort in patients undergoing robot-assisted laparoscopic prostatectomy. In general, the theme in the manuscript is a critical topic in our research field: anesthesiology and perioperative medicine. However, there are a few major concerns with this manuscript that need to be addressed by the authors.

  1. As the authors stated in the Discussion section, the main concern in this research is the lack of randomization. Non-randomized setting causes the presence of selection bias. Describe any efforts in your research to reduce selection bias.

According to the reviewer’s comment, in data analyses, we used multivariable modelling to adjust for between-patient differences in pre-operative and intra-operative characteristics. More specifically, we entered age, BMI, ASA score as well as all intra-operative variables in the multivariable models. This was specified in the methods section (see 2.3 Statistical analysis). Results section was modified according to the new findings.

  1. The authors showed the primary outcome, the chronological changes in the severity of bladder discomfort, in Figure 1. As the authors stated in the Methods section, the severity of the discomfort is an ordinal variable. Therefore, the authors should NOT use repeated measures ANOVA in the statistic. Also, the use of means of the ordinal variable is wrong. 

We agree with the referee that the calculation of CRBD mean values and ANOVA are not appropriate since CRBD is an ordinal variable, as it is stated in the Methods section. We therefore used ordinal logistic regression in order to estimate the effect of different analgesic treatments. Results are expressed in terms of regression coefficients, which estimate the difference between treatment groups in the log odds of being in an upper category of the dependent variable. This was specified in the methods section (see 2.3 Statistical analysis). The analysis was supported by a statistician (see acknowledgements).

  1. In addition, in modern statistics, the application of ANOVA is not appropriate for repeated measurements. Many researchers recommend the use of generalized linear mixed modeling for longitudinal data. Please carefully read two excellent reviews: Anesth Analg 2018;127:569 (PMID29905618) and Neuron 2022;110:21 (PMID34784504). Try to perform GLMM analysis again.  

We thank the referee for this comment and for suggesting these interesting reviews. As reported in the Methods section, we performed a multilevel ordinal logistic regression for the primary outcome (CRBD). For NRS we performed a multilevel linear regression, whereas for PONV we performed a multilevel logistic regression.

In all multilevel multivariable models, we used fixed effects for treatment groups, patients’ pre- and intra-operative characteristics and time, whereas we used random effects for patients. We also entered an interaction term between treatment groups and time in order to assess whether the effect of treatment changed with time. The covariance matrix was left unstructured. This was specified in the methods section (see 2.3 Statistical analysis).

Also, the reviewer recommends changing several minor points as described below.

  1. The explanation of the three treatment methods (IM, TAP, TI) is abrupt and confuses the reader a little. The author should explain them more carefully and in detail.

According to the reviewer’s comment, the following sentence was addded: “Drugs and techniques used to treat perioperative pain are often used even for this uncomfortable condition [2].”

  1. Statistical analysis. The authors should explain in more detail how the study size was determined. 

According to the reviewer’s comment, this poin was clarified as following: “As foreseen by a rule of the thumb on pilot studies [12], we decided to enroll 11 patients per group allowing for a 10% dropout, since 30 patients may meet the threshold for a suf-ficiently precise estimate of the variance of CRBD change.” The followign sentence was added: “Billingham, S.A.; Whitehead, A.L.; Julious, S.A. An audit of sample sizes for pilot and feasibility trials being undertaken in the United Kingdom registered in the United Kingdom Clinical Research Network database. BMC Med Res Methodol. 2013;13:104. doi: 10.1186/1471-2288-13-104.”

  1. To help the reader's understanding, the authors should round off data to the closest whole number. 

Results section has been revised according to findings of new analyses and following this reviewer’s advice. The two Figures have been replaced by two new tables with results of GLMM analysis. Summary was revised accordingly.

  1. Spelling and grammatical errors, particularly in the use of capital letters, should be corrected. The authors should request an English proofreading service to check the paper. 

According to the reviewer’s comment, the whole manuscript has been revised by a native English speaker (see acknowledgements).

Reviewer 2 Report

Thank you very much for giving me the opportunity to review this work. The authors investigated the effect of intrathecal morphine, TAP block, and tramadol infusion on catheter-related bladder discomfort in patients undergoing RALP, and found that intrathecal morphine was superior to other methods. CRBD is a highly uncomfortable condition, and the reviewer believe that its prevention is an important research topic.

Critiques:

  1. The reviewer does not believe that TAP block is effective for CRBD because TAP block is a method of blocking nerves distributed in the anterior abdominal wall; the authors should provide a rationale for believing that TAP block may be effective for CRBD.
  2. The authors compared intraoperative fentanyl doses as a secondary outcome. Methods states that intraoperative fentanyl dosing was based on heart rate and blood pressure fluctuations, but if there was a threshold for fentanyl dosing, it should be stated.
  3. Table 1. Please check PONV T3 in TAP group: “0711”.

The authors state that there is only one previous study that examined CRBD in the setting of RALP, but are there any previous studies in other settings?

  1. In addition to the non-randomized design, the small sample size is one of the limitations of this study. The reviewer believes that the sample size of this study does not allow for comparison of the frequency of side effects such as PONV.
  2. Even if intrathecal morphine is effective in preventing CRBD, the reviewer wonders whether the benefits outweigh the risks, since subarachnoid morphine administration carries the risk of rare but serious complications such as epidural hematoma. What are the authors' opinions on the risk-benefit balance of subarachnoid morphine administration? How many patients in the other groups complained of severe CRBD?

Author Response

Responses to Reviewer 2

Thank you very much for giving me the opportunity to review this work. The authors investigated the effect of intrathecal morphine, TAP block, and tramadol infusion on catheter-related bladder discomfort in patients undergoing RALP, and found that intrathecal morphine was superior to other methods. CRBD is a highly uncomfortable condition, and the reviewer believe that its prevention is an important research topic.

Critiques:

  1. The reviewer does not believe that TAP block is effective for CRBD because TAP block is a method of blocking nerves distributed in the anterior abdominal wall; the authors should provide a rationale for believing that TAP block may be effective for CRBD.

In order to clarify this point, we added the following sentence in the discussion section: “In this study, TAP block was not superior to tramadol infusion in CRBD treatment. The rationale behind the inclusion of the TAP block as comparative technique was based on the fact that performing the posterior approach of the TAP block, as done in this study, lo-cal anesthetic should extend throughout posterior medial pathway to the paravertebral region from T5 to L1 level, resulting in a transient paravertebral blockade. Paravertebral space contains spinal nerves as well as the dorsal and ventral branches of the spinal roots, and sympathetic chain [17]. Therefore, application of anesthetics into this space may re-sult in sensory and sympathetic block with consistent visceral analgesic effects.”The following reference was added: Carney J., Finnerty O., Rauf J., Bergin D., Laffey J. G., Mc Donnell J. G. Studies on the spread of local anaesthetic solution in transversus abdominis plane blocks. Anaesthesia. 2011;66(11):1023–1030. doi: 10.1111/j.1365-2044.2011.06855.x.

  1. The authors compared intraoperative fentanyl doses as a secondary outcome. Methods states that intraoperative fentanyl dosing was based on heart rate and blood pressure fluctuations, but if there was a threshold for fentanyl dosing, it should be stated.

According to reviewer’s comment, this issue was clarified as following “The threshold for fentanyl dosing was based on drug data sheet (600 mcg).”

  1. Table 1. Please check PONV T3 in TAP group: “0711”.

Revised.

  1. The authors state that there is only one previous study that examined CRBD in the setting of RALP, but are there any previous studies in other settings?

According to reviewer’s comment, we added in the discussion section the following sentence: “In previous studies conducted in other surgical settings, tramadol has been used in the control group to test the efficacy of other therapeutic strategies for CRBD [15,16].” The following reference was added: Lin F, Shao K, Pan W, Liang D, Zhao Z, Yuan J, Wang J, Lv Y. Comparison between Tramadol and Butorphanol for Treating Postoperative Catheter-Related Bladder Discomfort: A Randomized Controlled Trial. Evid Based Complement Alternat Med. 2021;2021:6002059. doi: 10.1155/2021/6002059.

  1. In addition to the non-randomized design, the small sample size is one of the limitations of this study. The reviewer believes that the sample size of this study does not allow for comparison of the frequency of side effects such as PONV.

We would to thank the reviewer for this comment. In this regard, we added the following sentence in the discussion section: “Another limitation is the small sample size that did not allow for appropriate comparison of the frequency of side effects such as PONV.”

  1. Even if intrathecal morphine is effective in preventing CRBD, the reviewer wonders whether the benefits outweigh the risks, since subarachnoid morphine administration carries the risk of rare but serious complications such as epidural hematoma. What are the authors' opinions on the risk-benefit balance of subarachnoid morphine administration? How many patients in the other groups complained of severe CRBD?

According to reviewer’s comment, we added in the discussion section the following sentence: “Since subarachnoid morphine administration carries the risk of rare but serious complications such as epidural hematoma, the risk-benefit balance should be evaluated in the single patients and based on the concomitant need to counteract pain in a stronger manner.

Round 2

Reviewer 1 Report

Authors do not fully reflect the reviewers' suggestions to the manuscript, but I think it has a form as a paper.